# Flexible coding of time or distance in hippocampal cells

Shai Abramson[1], Benjamin J Kraus[2], John A White[3], Michael E Hasselmo[2], Dori Derdikman[1]*[†], Genela Morris[1,4]*[†]

[1]Rappaport Faculty of Medicine and Research Institute, Technion - Israel Institute of Technology, Haifa, Israel; [2]Center for Memory and Brain, Boston University, Boston, United States; [3]Department of Biomedical Engineering, Boston University, Boston, United States; [4]Tel Aviv Sourasky Medical Center, Tel Aviv, Israel

**Abstract** Analysis of neuronal activity in the hippocampus of behaving animals has revealed cells acting as 'Time Cells', which exhibit selective spiking patterns at specific time intervals since a triggering event, and 'Distance Cells', which encode the traversal of specific distances. Other neurons exhibit a combination of these features, alongside place selectivity. This study aims to investigate how the task performed by animals during recording sessions influences the formation of these representations. We analyzed data from a treadmill running study conducted by Kraus et al., 2013, in which rats were trained to run at different velocities. The rats were recorded in two trial contexts: a 'fixed time' condition, where the animal ran on the treadmill for a predetermined duration before proceeding, and a 'fixed distance' condition, where the animal ran a specific distance on the treadmill. Our findings indicate that the type of experimental condition significantly influenced the encoding of hippocampal cells. Specifically, distance-encoding cells dominated in fixed-distance experiments, whereas time-encoding cells dominated in fixed-time experiments. These results underscore the flexible coding capabilities of the hippocampus, which are shaped by over-representation of salient variables associated with reward conditions.

*For correspondence:
derdik@technion.ac.il (DD);
genelam@tlvmc.gov.il (GM)

[†]These authors contributed
equally to this work

Competing interest: The authors declare that no competing interests exist.

## Editor's evaluation

The manuscript is a new analysis of previously published data from experiments in which rats ran on a treadmill in either fixed-time or fixed-distance trials. The valuable results provide convincing evidence to demonstrate that time and distance cells are more common in fixed-time and fixed-distance trials, respectively. These findings suggest that the hippocampus flexibly shifts between representing variables depending on their relevance.

## Introduction

The hippocampus plays an important role in spatial processing and episodic memory (*Andersen et al., 2006*; *Tulving, 2002*). Spatial processing and navigation are supported by spatially tuned cells throughout the hippocampal formation, such as place cells in the hippocampus, which sparsely encode location within an environment (*O'Keefe and Dostrovsky, 1971*; *Muller and Kubie, 1987*). Subsequent discovery of time cells in the hippocampus (*Kraus et al., 2013*; *Pastalkova et al., 2008*; *MacDonald et al., 2011*; *Tsao et al., 2018*; *Rueckemann et al., 2021*), which encode time within an episode, suggests that the latter may contribute to the building blocks of episodic memory formation. Time cells and place cells share many physiological properties, pointing to a unifying concept of the role of the hippocampus in encoding features required to organize relevant information. We asked whether the encoding of hippocampal neurons is flexible, capable of changing according to

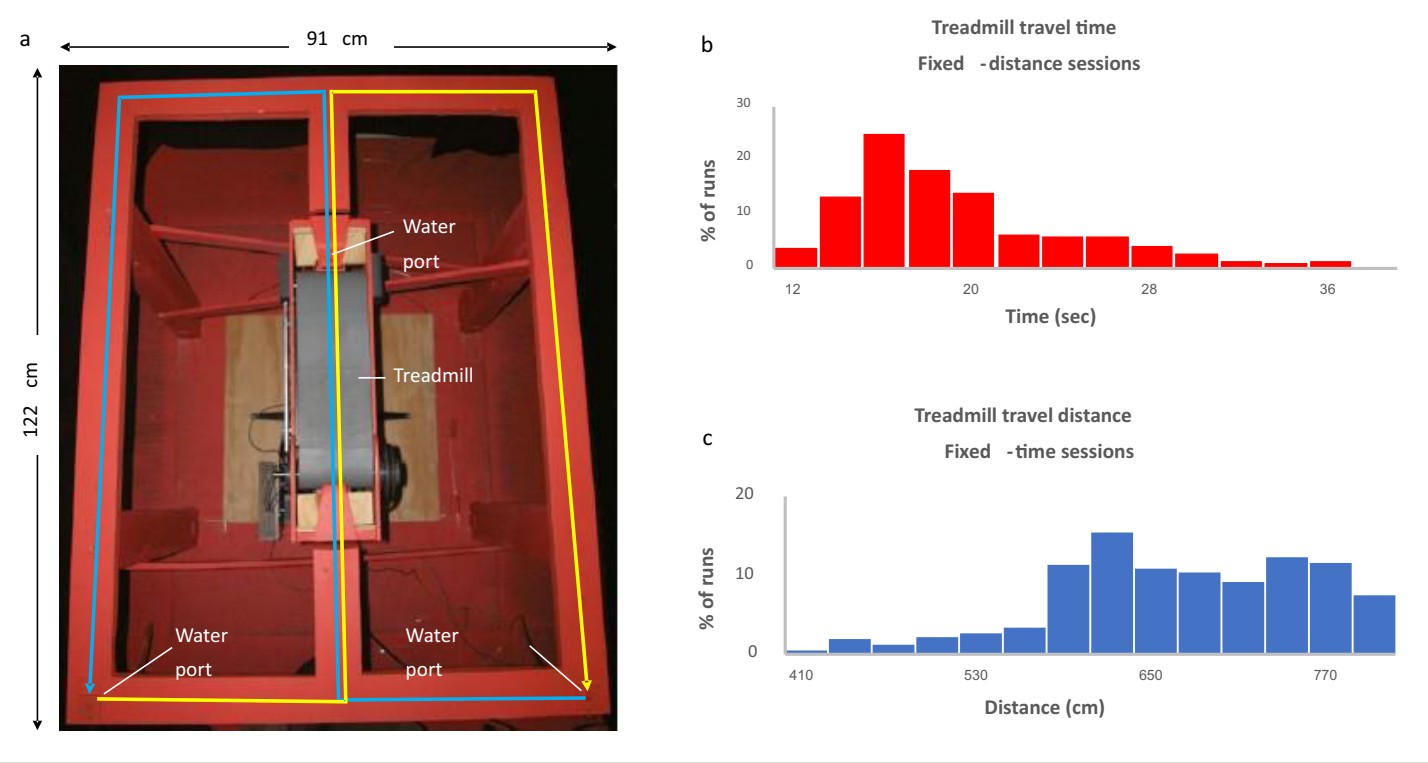

**Figure 1.** Experimental setup. (**a**) The figure-eight maze with treadmill (gray belt) located in the central stem. Water ports are located near the treadmill and at the two lower corners of the maze. Blue line indicates right-to-left alternation; yellow line indicates left-to-right alternation. (**b, c**) Distribution of the fixed-time sessions treadmill travel times (**b**) and the fixed-distance sessions treadmill travel distances (**c**).

behavioral context and task demand. We used previously published data by *Kraus et al., 2013*, from an experiment which sought to resolve an inherent ambiguity in the interpretation of time cells. Time cells were initially reported in animals running on a running wheel without control of velocity (*Pastalkova et al., 2008*, although time cells were also reported for immobile rats *MacDonald et al., 2011*). This led to a potential ambiguity between encoding of time and of distance, due to the fact that, in fixed velocities, distance may be encoded by integration of time. *Kraus et al., 2013* varied the velocity of rats running in place on a treadmill, and found subpopulations of hippocampal cells that encoded time, other cells that encoded path-integrated distance and additional cells that encoded both time and distance. These experiments were composed of two types of recording sessions. In one type of session, in all the trials the running duration remained constant at different velocities, whereas in the second type, the treadmill runs accumulated up to a constant distance, at different velocities. We hypothesized that in this experiment, the task demand (i.e. constant time vs. constant distance) determined the type of activity exhibited in the corresponding session. We re-analyzed the data according to the type of behavioral session and found a direct relation between the class of most active cells and the type of session in which they were recorded. In sessions in which the rats ran for a fixed time, the cells' population was dominated by time-encoding cells, while in sessions where they ran for a fixed distance, the population was dominated by distance-encoding cells.

## Results

To examine the dependence of hippocampal coding on task demand, we analyzed data based on experiments by *Kraus et al., 2013*, which aimed to differentiate between cells encoding time and cells encoding distance in the hippocampus. In these experiments, six rats were trained to run on a treadmill in the central stem of a figure-8 maze (*Figure 1a*). The rats were provided with a small water

reward prior to the initiation of the treadmill session and upon its cessation, thereby conditioning them to maintain their snouts positioned at the water port throughout the duration of the treadmill run, and "clamping" their behavior and spatial position. In each session, consisting of 31–57 runs, the treadmill was operated either for a fixed time or for a fixed distance, where in each run the velocity was set to a randomly chosen speed in the range of 35–49 cm/s (*Figure 1b and c*). The rats were forced to alternate their post-treadmill turns between right and left. Three of the six rats were trained and recorded exclusively in fixed-distance or fixed-time sessions, while the remaining three rats were trained and recorded in sessions of both types.

Kraus et al. reported that some cells preferentially encoded the distance the rat had run on the treadmill, while other cells preferentially encoded the time from the start of the treadmill movement. We hypothesized that the type of task employed in each session (i.e. fixed-time vs. fixed-distance) would determine the encoding of the neurons (i.e. time-based vs. distance-based). We therefore analyzed the cells on a run-by-run basis, as follows: For each neuron, we defined its response in each run according to the onset of peak firing (see Materials and methods), and examined its relation to the treadmill's velocity. We classified time-encoding cells as those, in which the response did not systematically depend on the treadmill velocity but instead fired at a fixed time after the initiation of treadmill running. We classified distance-encoding cells as those, in which the onset time was proportional to the treadmill velocity. To examine this classification, we determined the firing onset of each

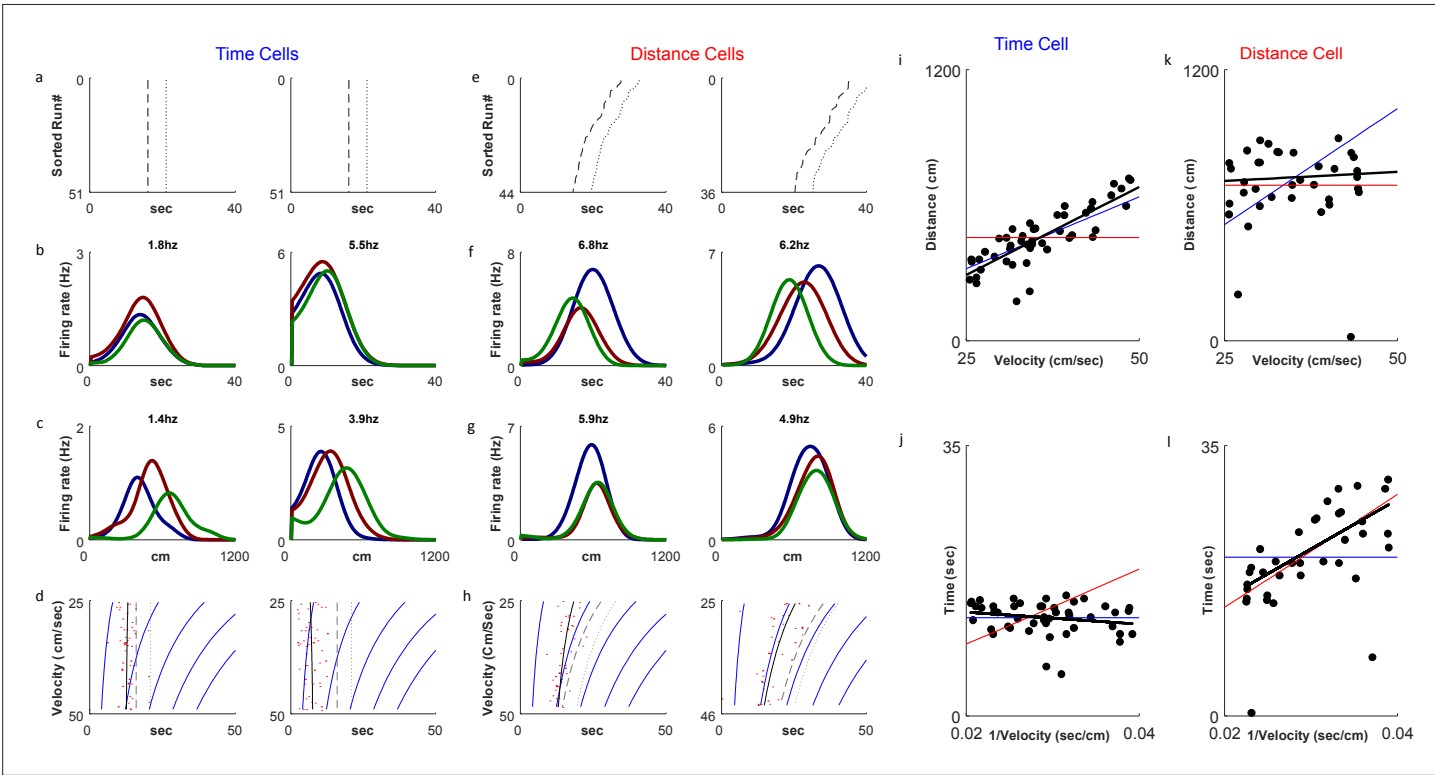

**Figure 2.** Distance and time cells coding. (**a–h**) Examples of two time-coding cells (columns 1–2) and two distance-coding cells (columns 3–4). Row a depicts neural firing as a function of the distance the animal traveled, sorted by the runs' velocities. The colors represent three velocity groups for which the tuning curves, by time or distance, are presented in rows b and c, respectively. Row d shows the onsets of each run (red dots) and their linear fit (black curve) to the relation $S = m * v + n$ for the time cells and $T = k * \frac{1}{V} + q$ for the distance cells. The dashed curve represents the end-of-run time, and the dotted curve represents the end of the analyzed period (treadmill stop time, plus 5 seconds). The blue curves are equi-distance points in time. A black curve (the linear fit) which is parallel to the equi-distance curves demonstrates a cell with strong distance coding. (**i–l**) Examples of the analysis of time (**i, j**) and distance (**k, l**) encoding cells. Top row plots depict distance vs. velocity and bottom row plots depict onset vs. 1/velocity. Red line represents an ideal Distance Cell, based on the average distance traveled until the onset time. Blue line represents an ideal Time Cell, based on the average time of the onset and the black line is the linear fit. The closer the slope of the black line is to that of the red line, relative to slope of the blue line, the more the cell encodes distance-encoding, while if the slope is closer to the slope of the blue line, the is more it is time-encoding.

The online version of this article includes the following figure supplement(s) for figure 2:

**Figure supplement 1.** Additional examples of cells showing time coding (**a–d**) and distance encoding (**e–h**).

cell in each run and determined the cell's properties according to three metrics (see Materials and methods section).

We defined a metric, $CellType = \frac{Var(distance) - Var(time)}{Var(distance) + Var(Time)}$ , based on the distance and time variances (see Analysis Methods). We classified cells with positive CellType, where the distance variance is greater than the time variance as time cells, and negative CellType, where the time variance is greater than the distance variance, as distance cells. For an ideal time cell, CellType = 1 and for an ideal distance cell, CellType=-1.

Of 930 cells recorded we analyzed 679 cells with at least 10 runs showing firing peaks greater than 0.5 Hz. Only cells with peak firing rates occurring during the treadmill run were included in the analysis. As previously reported in *Kraus et al., 2013*, we observed both distance cells, showing a response at constant distances the animal traveled on the treadmill, and time cells, showing a response at constant times from the treadmill start (*Figure 2*). In line with our hypothesis, there was a clear relation between the types of experiment and the distribution of time coding and distance coding neurons. In fixed-distance sessions, the neurons exhibited a significant majority (67%) of distance cells. By contrast, in fixed-time sessions time encoding cells dominated (68%). Of the 444 neurons recorded in fixed-distance sessions, the CellType index classified 298 cells as distance cells and 146 as time cells (*Figure 3a*). Conversely, in the fixed-time sessions 76 of 235 neurons were classified by this index as distance cells and 159 of 235 were classified as time cells. The relation between the cell type and the experiment type is significant ($\chi^2$(1)=75.1, p>>0.001 for the total cell population). These proportions were maintained when classifying by other metrics and when using the peak of firing instead of the onset of the response (see supplementary Methods and *Figure 3—figure supplement 1*). In 5 out of 6 animals, the cells' encoding depended on the session type (*Figure 3—figure supplement 2*). ($\chi^2$(1)>12, p<<0.001 for 4 animals, $\chi^2$(1)=5.7, p<0.02 for one animal), except in one animal ($\chi^2$(1)=2.38, p=0.12).

We then checked the Receiver Operating Characteristics (ROC) graph of the CellType metric (*Figure 2d*), using discriminating thresholds in the range of [–1,+1]. The ROC plots the True Positive Rate (TPR) defined as the percentage of cell classified as distance cells in the fixed-distance session, against the False Positive Rate (FPR) defined as the percentage of cells classified as distance cells in the fixed-timed sessions. We found that the optimal threshold (maximum Youden index) for classification of a session based on the CellType metric, was 0. This threshold classifies 67% of the cells as Distance in the Distance sessions and only 32% on the Time sessions. Accordingly, we classified a cell as a time-cell if CellType >0 and as a distance-cell if CellType <0.

To assess the power of the statistics, we compared the results to a distribution generated from shuffled session types (*Figure 3c*). In order to mitigate any potential biases in this distribution, we truncated all data to a common duration of 16 s, which represents the shortest duration of a treadmill run across all sessions. These results indicate that the dimension the cells encode (Time vs. Distance) is related to the session type (fixed time vs. fixed distance).

## Discussion

Classifying neuronal activity according to either time or distance revealed that the hippocampal population encoding strongly registered with the features of the experimental task. In experiments where the treadmill running-time was fixed, the majority of cells encoded a given time from treadmill onset. In contrast, in experiments where the treadmill running-distance was fixed, the majority of cells encoded a specific accumulated distance from treadmill onset. It is worth noting that accumulated time in fixed-time experiments and accumulated distance in fixed-distance experiments may be used as predictors for the progress of the rat towards anticipated reward, which is given at the end of the treadmill run (*Whittington et al., 2020*; *Stachenfeld et al., 2017*). As noted previously in *Kraus et al., 2013* the same cells, which showed distance-encoding and time-encoding properties in the treadmill, were often selective to places outside of the treadmill as well. To summarize, CA1 pyramidal cells can encode location, distance, or time, depending on the conditions of the experiment or task demand.

Consistency with task demands has been repeatedly demonstrated in hippocampal recording for diverse parameter spaces, such as auditory linear frequency (*Aronov et al., 2017*), social mapping (*Omer et al., 2018*; *Schafer and Schiller, 2018*) or more abstract spaces (*Constantinescu et al., 2016*; *Retailleau and Morris, 2018*). How is task-relevant encoding achieved? The activity of place cells

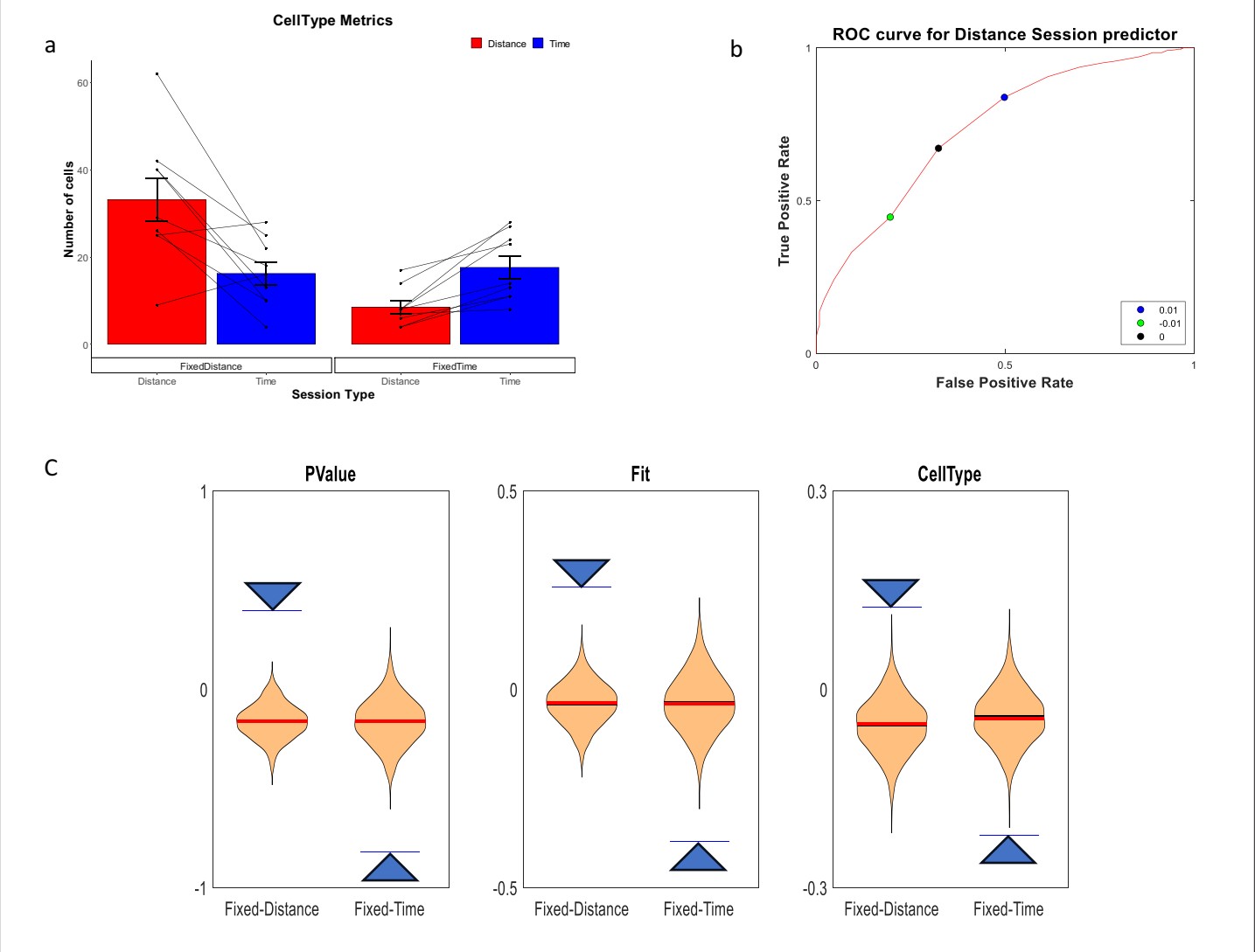

**Figure 3.** Distance and Time Cells classification. (**a**) Cells type classified by the CellType metric, averaged over all animals and trials, for fixed-distance and fixed-time experiments (n=18 experiments, mean ± SEM) p<<0.001 by Pearson's chi-squared test using two categories. Diagonal lines represent individual animals. (**b**) ROC curve (red) showing that the chosen discriminating threshold of 0 (black point) is optimal. The True Positive Rate (TPR) is the percentage of cells classified as distance cells on the fixed-distance session, while the False Positive Rate (FPR) is the percentage of cells classified as distance cells on the fixed-time sessions. (**c**) Shuffling distribution of the three metrics: CellType, FIT and P-Value. The type of experiment, either fixed-time or fixed-distance, was randomized 1000 times for each of the sessions. All experiments were truncated to 16 s in order to prevent biases. The vertical axis is the Time-Distance balance index (TDI), defined as (#DistanceCells-#TimeCells)/(#DistanceCells + TimeCells) and is between 0 and 1 if there are more distance cells than time cells and between –1 and 0 if there are more time cells than distance cells. The arrows are indicating the actual results which are significant compared to the shuffle distribution.

The online version of this article includes the following figure supplement(s) for figure 3:

**Figure supplement 1.** Additional metrics.

**Figure supplement 2.** The distribution of Time and Place cells, per animal, at various metrics, for Time and Distance sessions.

**Figure supplement 3.** Distribution of max and average firing rates for Time and Distance cells.

and grid cells is commonly modeled using continuous attractor networks (*Morris and Derdikman, 2023*; *Samsonovich and McNaughton, 1997*; *Burak and Fiete, 2009*; *Fuhs and Touretzky, 2006*; *Bush and Burgess, 2014*; *Hasselmo and Brandon, 2012*; *Tocker et al., 2015*; *Gu et al., 2018*; *Geiller et al., 2022*; *O'Keefe and Burgess, 1996*). Such networks may serve as a natural substrate for amplification of encoding of certain task features, at the expense of others, resulting in an over representation of the salient variables. Such over-representation may help the brain prioritize survival

and decision making. By allocating more resources to the salient stimuli, the brain enhances the ability to process and retain the important information.

Another possible mechanism for acquiring representations that are consistent with task structure involves an associative learning process. Such learning would strengthen all connections between cells that were active in a particular trial and weaken others, but ultimately only the connections between cells that are consistently co-active will be strengthened, while others will average out. Thus, in time-fixed experiments, the connections to cells that fire in a manner that is consistent with time would be strengthened, while in distance-fixed experiments connections to cells that are consistent with distance would gain strength. Consequently, those cells will gradually encode either distance or time, depending on the type of experiment.

Irrespective of the exact mechanism explaining the results of this study, the hippocampus is adaptive in its cells' encoding and seems to be capable to tune them to the parameters best describing the task.

## Materials and methods

We used the data provided by *Kraus et al., 2013*, containing the neurons firing times, the treadmill movement times and the treadmill velocity. The data was analyzed using custom Matlab scripts.

We divided the treadmill moving times into 100ms time bins (other bin resolutions between 50 ms and 500 ms were tested and provided similar results). Response onset for each neuron and run, was defined as the first bin, following at least 1 s of silence within a series of consecutive bins with firing activity which includes the peak of firing. The peak firing was defined as the bin with the highest value within a run, however only cells with an average peak firing of at least 0.5 Hz were included. This methodology was applied consistently across all runs within a session. We chose this approach to mitigate potential biases that could arise from firing rate peaks occurring near the end of the treadmill, which might have been truncated. Basing our analysis on the peak values instead yielded comparable results and levels of statistical significance (*Figure 3—figure supplement 1*).

Our classification method is based on the premise that for ideal time cells, firing (and hence onset time) should be independent of running speed. Conversely, an ideal distance cell would display firing onsets at times depending on the speed. We therefore performed a linear regression between the onset times ($T_i$) and the reciprocal velocity ($1/V_i$) as well as the onset distance ($S_i$) and the velocity ($V_i$), where $i$ is an index to a specific run on the treadmill, and extracted the slopes ($m$ and $k$) and offsets ($n$ and $q$), as described in *Equations 1 and 2*. Consequently, in the case of a time-encoding cell, the velocity dependent term $k * \frac{1}{V_i}$ would be small in comparison to the constant $q$, while for a distance encoding cell, the slope $k$ would approximate the estimated encoded distance. Similarly, in accordance with the relation stated in *Equation 2*, an ideal distance encoding cell would exhibit a small velocity dependent term $m * V_i$ compared to the constant $n$, while a time encoding cell would have an estimated encoded time equivalent to the slope $m$.

$$T_i = k * \frac{1}{V_i} + q \tag{1}$$

$$S_i = m * V_i + n \tag{2}$$

The CellType metric utilizes the variances of the onset times $(T_i - \bar{T})^2$, where the average onset time is computed across all runs within the session, and the onset distances $(S_i - \bar{S})^2$, where the average onset distance is calculated across all runs within the session.

$$CellType\left(V_i, S_i, T_i\right) = \frac{\sum_i V_i * (S_i - \bar{S})^2 - \sum_i (T_i - \bar{T})^2}{\sum_i V_i * (S_i - \bar{S})^2 + \sum_i (T_i - \bar{T})^2} \tag{3}$$

CellType is in the range of –1 to +1. For an ideal time-encoding cell, the onset variance $\sum_i (T_i - \bar{T})^2 = 0$, and hence CellType = 1. For an ideal distance encoding cell, the distance variance (multiplied by the respective velocity in order to match units) $V_i * (S_i - \bar{S})^2 = 0$, and hence CellType=-1.

Additional metrics defined and used for classifying the cells encoding:
The "FIT" metric is defined as follows:

$$\text{Fit}\left(m,k,\bar{T},\bar{S}\right) = \begin{cases} -1 & 0.5 < \dfrac{m}{\bar{T}} \\ 1 & 0.5 < \dfrac{k}{\bar{S}} \\ 0 & otherwise \end{cases} \tag{4}$$

Where $m$ and $k$ are the linear fit slope coefficients (from *Equations 1 and 2*), $\bar{T}$ is the average firing onset time and $\bar{S}$ is the average distance the animal traveled until the onset. Fit is –1 for a distance cell and 1 for a time cell.

The 'p-value' metric is defined as follows:

$$P_{Value}\left(V_i, S_i, T_i\right) = \begin{cases} -1 & \text{p(no linear relation between the distance and velocity)} < 0.05 \\ 1 & \text{p(no linear relation between the onset time and the reciprocal velocity)} < 0.05 \\ 0 & otherwise \end{cases}$$

A stricter metric, utilized the statistical significance of the linearity in *Equations 1 and 2*, through F-statistics. We classified a cell as distance encoding if the null hypothesis that there is no linear relation between the distance and velocity was rejected with $p < 0.05$. We classified a cell as time encoding if the null hypothesis that there is no linear relation between the onset and the reciprocal velocity was rejected with $p < 0.05$.

Results using these metrics are shown in *Figure 3—figure supplement 1*.

To ensure the activity peak is not missed, we extended the analysis to 5 s past the treadmill stop time. Otherwise, if a cell activity is concentrated towards the treadmill stop, the calculated onset may be influenced by the truncated activity time and show a false relation of the cell type activity to the time or distance. Moreover, since the truncated data time relates to the experiment type, whether time-fixed or distance-fixed, this could create a false bias of such a relation.

The relation between the type of cell classified in the above metrics and the session type was then tested by Pearson's chi-square using two categories. The expected distribution of the cells was calculated based on the total number of cells, of each type, out of total cells number, in all sessions. The null hypothesis was defined as no dependency of the cells type distribution on the session type (either fixed-time or fixed-distance). On the per-animal analysis, for animals that were recorded only at a single type of session, we used the distribution of the cell types according to their distribution in all animals' cells population.

We conducted additional analysis to explore potential relationships between the firing rates and the encoding properties of the cells. Our findings revealed that the distributions of peak firing rates and average firing rates, for time cells and distance cells, were similar (see *Figure 3—figure supplement 3*).

---

## Additional information

### Funding

| Funder | Grant reference number | Author |
|---|---|---|
| Israel Science Foundation | 2183/21 | Dori Derdikman |
| Binational Science Foundation (BSF)-NIH CRCNS | BSF:2019807 (NIH: 1R01 MH125544-01 ) | Dori Derdikman |
| Prince Center for the Aging Brain | | Dori Derdikman |
| Israel Science Foundation | 3139/22 | Genela Morris |

The funders had no role in study design, data collection and interpretation, or the decision to submit the work for publication.

## Author contributions
Shai Abramson, Conceptualization, Software, Investigation, Methodology, Writing - original draft; Benjamin J Kraus, John A White, Conceptualization, Data curation, Methodology, Writing – review and editing; Michael E Hasselmo, Conceptualization, Methodology, Writing – review and editing; Dori Derdikman, Genela Morris, Conceptualization, Supervision, Investigation, Methodology, Writing – review and editing

## Author ORCIDs
Shai Abramson ⬤ http://orcid.org/0000-0002-5437-7992
John A White ⬤ http://orcid.org/0000-0003-1073-2638
Dori Derdikman ⬤ http://orcid.org/0000-0003-3677-6321
Genela Morris ⬤ http://orcid.org/0000-0002-5417-8977

## Ethics
The data in this paper is based on the study in a previous paper (Kraus et al., 2013), and no experiments have been thus performed specifically for this study.

## Decision letter and Author response
Decision letter https://doi.org/10.7554/eLife.83930.sa1
Author response https://doi.org/10.7554/eLife.83930.sa2

## Additional files

### Supplementary files
• MDAR checklist

### Data availability
The current manuscript is a re-analysis of data collected for a previously published paper (*Kraus et al., 2013*). Data used in this paper is available as Matlab files on Dryad: https://doi.org/10.5061/dryad.ngf1vhhxp. Matlab code used for the analysis in the paper is available on https://github.com/derdikman/Abramson_code (copy archived at *Derdikman, 2023*).

The following dataset was generated:

| Author(s) | Year | Dataset title | Dataset URL | Database and Identifier |
|---|---|---|---|---|
| Abramson S, Kraus BJ, White JA, Hasselmo ME, Morris G, Derdikman D | 2023 | Data for Time or distance: predictive coding of hippocampal cells | https://doi.org/10.5061/dryad.ngf1vhhxp | Dryad Digital Repository, 10.5061/dryad.ngf1vhhxp |

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
