## [Editor Report]

The manuscript is a new analysis of previously published data from experiments in which rats ran on a treadmill in either fixed-time or fixed-distance trials. The valuable results provide convincing evidence to demonstrate that time and distance cells are more common in fixed-time and fixed-distance trials, respectively. These findings suggest that the hippocampus flexibly shifts between representing variables depending on their relevance.

---

## [Decision Letter]

**Decision letter after peer review:**

Thank you for submitting your article "Time or distance: predictive coding of Hippocampal cells" for consideration by *eLife*. Your article has been reviewed by 3 peer reviewers, and the evaluation has been overseen by a Reviewing Editor and Laura Colgin as the Senior Editor. The reviewers have opted to remain anonymous.

Essential revisions:

1) Reviewers agreed that the authors have not been precise and clear enough about what they are referring to as a predictive code and the source of its importance. That is, authors need to provide a properly-referenced definition of what is meant by predictive coding (including to what degree it is or is not distinct from flexibility of representation). Also, authors should provide a summary of why evidence for predictive coding is interesting and important (with respect to hippocampal function). Otherwise, it is not possible to assess why the described findings provide evidence for predictive coding. If the authors mean to use the term "predictive coding" in the same manner in which it has been used in prior publications, then more analysis is needed to support the claim (see individual reviews below for details). However, the main point of the paper was agreed to be the flexible representation of time vs. distance across two tasks, and this finding was viewed as interesting on its own. Therefore, if they prefer, authors can choose to remove the claims about predictive coding, rather than performing more analyses to substantiate the claims related to predictive coding.

2) Other analyses and points of clarification are suggested to strengthen the flexible representation of time vs. distance results. See individual reviews below for details.

*Reviewer #1 (Recommendations for the authors):*

In this manuscript the authors examine whether place cells (cells that encode specific location on an environment) or "elapsed time cells" (cells that encode total time elapsed on a track) depend on the type of task that the animal is doing. The authors show that in tasks were fixed-distance travel is important, there is a bias for distance-encoding cells and on fixed-time experiments, there is a bias for time-encoding cells. The authors conclude that this indicates that the hippocampal code has a predictive component, as it generates firing patterns that tile the relevant features of each of the tasks respectively. Overall, the analyses are well done, and the results are clear. My main concern is that even if the hippocampal code generates responses that match the most needed variable for each task, it also generates responses that match the alternative variables. For example, in the elapsed time task, there are also place cells and in the fixed-distance travel there are also cells that encode other features. This, rather than a predictive code, can be a regular sample of the environment with an overrepresentation of the more salient variable that animals need to get in order to collect rewards. In addition, the analysis provided in the manuscript are rather simple, and better controls could be provided. Improving the analytical quantification of the results is necessary to support the main claim.

– What is the relationship of each type of cell with the speed of the animal?

– What is the relationship with the n of trial that the animal has run (first 10 trials, last 10 trials)?

– What is the average firing rate of each neuron? Is there any relationship between intrinsic firing rate and the type of coding that the cell develops in each task?

– What is the relation of the units of each type with LFP features (theta phase, ripple recruitment)?

*Reviewer #2 (Recommendations for the authors):*

I have a couple of comments that I think should be addressed to make the manuscript stronger.

1. It is very unusual to define timing of a cell as the earliest onset of timing prior to the peak. It seems arbitrary, and also not very robust measure: it effectively uses only the earliest spikes fired by the cell on a given trial, instead of using all of the spikes. This measure should therefore be highly susceptible to noise. It's unclear why the authors didn't do something more simple and standard, such as measuring the actual peak (or center of mass) of the firing. Alternatively, they could've fit some template to the spikes and allowed this template to expand and contract to determine whether the firing is more consistent with distance or time coding. If the authors are going to stick with the method they are currently using, I suggest adding a clear justification about why it is the best method to use for distinguishing distance cells from time cells.

2. To be completely honest, I could not understand the model presented in the Discussion, in spite of multiple re-readings. I think the description suffers from being very brief and from using shortcuts in language that are hard to interpret. E.g. phrases like "connections that were active" (how can a connection be active or inactive), "connections that are consistent with time" (what does it mean for a connection to be consistent with something), "signal … coinciding with cells" (what does it mean for a signal to coincide with a cell), and "cells will gradually encode" (what exactly is changing gradually)? I suggest substantially expanding the explanation and using concrete, mechanistic language. Figure 4 that illustrates the model is visually stunning, but is also very hard to interpret and doesn't seem to contribute much to the explanation.

The best option would be to run an actual simulation that produces time or distance encoding in different situations. But I don't think this is required; a more clear explanation should be enough.

*Reviewer #3 (Recommendations for the authors):*

The main hypothesis test (session type independent of cell responses) is parametric and lacks a measure of strength of effect. A crude measure of strength of effect combined with a more robust hypothesis test would be provided by shuffling session type labels. Even a simple comparison of the proportion in the shuffled data gives a rough estimate of the strength of the effect.

Please motivate the classifiers and their statistical assumptions scientifically. In particular, what alternatives do they have power against, and why are these power profiles of interest scientifically. I didn't understand why you were testing for deviations from linearity to build the third classifier. Was there a motivation for that?

The abstract, for example, spends too much time replicating the Kraus paper. More time should be spent explaining why the results imply hippocampal cells encode prediction, as well as motivating why that's important or potentially important. Related to this, the reward hypothesis is not developed strongly enough.

"encoding dimensions required in order to organize relevant information." What does this mean? Perhaps simplifying it would be helpful. Does dimension mean features of sensory experience?'

I felt the final paragraph was important but not precise enough and therefore hard to parse.

The explicit structure of reward and its relationship to the main conjecture is not really discussed. This seems crucial. I'd prefer to have it upfront in the manuscript, rather than mentioning it obliquely at the beginning and coming back to it at the end.

Are we evaluating our results on the same data we are using to fit the classifiers? And does that matter?

---

## [Author Response]

Essential revisions:1) Reviewers agreed that the authors have not been precise and clear enough about what they are referring to as a predictive code and the source of its importance. That is, authors need to provide a properly-referenced definition of what is meant by predictive coding (including to what degree it is or is not distinct from flexibility of representation). Also, authors should provide a summary of why evidence for predictive coding is interesting and important (with respect to hippocampal function). Otherwise, it is not possible to assess why the described findings provide evidence for predictive coding. If the authors mean to use the term "predictive coding" in the same manner in which it has been used in prior publications, then more analysis is needed to support the claim (see individual reviews below for details). However, the main point of the paper was agreed to be the flexible representation of time vs. distance across two tasks, and this finding was viewed as interesting on its own. Therefore, if they prefer, authors can choose to remove the claims about predictive coding, rather than performing more analyses to substantiate the claims related to predictive coding.

The reviewers’ comments are appreciated and were fully implemented. We have thus removed the claims about predictive coding and changed the title to "Flexible coding of time or distance in hippocampal cells" (page 1 line 1).

Reviewer #1 (Recommendations for the authors):In this manuscript the authors examine whether place cells (cells that encode specific location on an environment) or "elapsed time cells" (cells that encode total time elapsed on a track) depend on the type of task that the animal is doing. The authors show that in tasks were fixed-distance travel is important, there is a bias for distance-encoding cells and on fixed-time experiments, there is a bias for time-encoding cells. The authors conclude that this indicates that the hippocampal code has a predictive component, as it generates firing patterns that tile the relevant features of each of the tasks respectively. Overall, the analyses are well done, and the results are clear. My main concern is that even if the hippocampal code generates responses that match the most needed variable for each task, it also generates responses that match the alternative variables. For example, in the elapsed time task, there are also place cells and in the fixed-distance travel there are also cells that encode other features. This, rather than a predictive code, can be a regular sample of the environment with an overrepresentation of the more salient variable that animals need to get in order to collect rewards.

We thank the reviewer for suggesting the overrepresentation interpretation as an alternative. The following was added:

“These results underscore the flexible coding capabilities of the hippocampus, which are shaped by over-representation of salient variables associated with reward conditions.” (page 1 line 23)

“Such over-representation may help the brain prioritize survival and decision making. By allocating more resources to the salient stimuli, the brain enhances the ability to process and retain the important information.” (page 5 line 18)

In addition, the analysis provided in the manuscript are rather simple, and better controls could be provided. Improving the analytical quantification of the results is necessary to support the main claim.

We added a shuffling analysis to assess the power of our results:

“To assess the power of the statistics we compared the results to a distribution generated from shuffled trial types (Figure 3c). In order to mitigate any potential biases in this distribution, we truncated all data to a common duration of 16 seconds, which represents the shortest duration of a treadmill run across all trials.” (page 4 line 24 and figure 3c)

Reviewer #2 (Recommendations for the authors):I have a couple of comments that I think should be addressed to make the manuscript stronger.1. It is very unusual to define timing of a cell as the earliest onset of timing prior to the peak. It seems arbitrary, and also not very robust measure: it effectively uses only the earliest spikes fired by the cell on a given trial, instead of using all of the spikes. This measure should therefore be highly susceptible to noise. It's unclear why the authors didn't do something more simple and standard, such as measuring the actual peak (or center of mass) of the firing. Alternatively, they could've fit some template to the spikes and allowed this template to expand and contract to determine whether the firing is more consistent with distance or time coding. If the authors are going to stick with the method they are currently using, I suggest adding a clear justification about why it is the best method to use for distinguishing distance cells from time cells.

We chose the onset rather than the peak since in some cases the firing field is truncated by the end of the run. We were therefore worried that choosing the peak might introduce a bias. Following the comment, we repeated the analysis in relation to the peak of firing of the cells. As we show in the figures added to the paper, this did not change our results.

The following passage was added to the paper:

“We chose this approach to mitigate potential biases that could arise from firing rate peaks occurring near the end of the treadmill, which might have been truncated. Basing our analysis on the peak values instead yielded comparable results and levels of statistical significance (figure 3—figure supplement 1).” (page 6 line 9)

2. To be completely honest, I could not understand the model presented in the Discussion, in spite of multiple re-readings. I think the description suffers from being very brief and from using shortcuts in language that are hard to interpret. E.g. phrases like "connections that were active" (how can a connection be active or inactive), "connections that are consistent with time" (what does it mean for a connection to be consistent with something), "signal … coinciding with cells" (what does it mean for a signal to coincide with a cell), and "cells will gradually encode" (what exactly is changing gradually)?

We agree with the reviewer that the model in its current form is not clear enough. Following this comment, and our realization that the model is still not fully characterized, we decided to omit the model part from the revised version of the paper.

I suggest substantially expanding the explanation and using concrete, mechanistic language. Figure 4 that illustrates the model is visually stunning, but is also very hard to interpret and doesn't seem to contribute much to the explanation.

Figure removed.

Reviewer #3 (Recommendations for the authors):The main hypothesis test (session type independent of cell responses) is parametric and lacks a measure of strength of effect. A crude measure of strength of effect combined with a more robust hypothesis test would be provided by shuffling session type labels. Even a simple comparison of the proportion in the shuffled data gives a rough estimate of the strength of the effect.

A shuffling analysis was added demonstrating that the results are highly significant compared to the shuffled distribution.

Please motivate the classifiers and their statistical assumptions scientifically. In particular, what alternatives do they have power against, and why are these power profiles of interest scientifically. I didn't understand why you were testing for deviations from linearity to build the third classifier. Was there a motivation for that?

The three metrics were designed to as different statistics describing the linear regression between the firing times (either onset or peak) and the velocity of the run.

a) The CellType metric is defined by the normalized ratio of the residuals’ sum of the firing onset times in each run under constant time vs. constant distance assumptions. It is designed to have values between -1 and 1, such that -1 is a pure time cell and +1 is a pure distance cell.

b) The Fit metric examines the ratio between the average onset time ***T*** and the slope of the distance vs. reciprocal velocity linear regression (m), and also the ratio between the average onset distance ***S*** and the slope of the time vs. velocity linear regression slope (q). If the ratio for onset time exceeds 0.5, the cell is classified as a distance cell, indicating a strong association between time and reciprocal velocity. Conversely, if the ratio for onset distance exceeds 0.5, the cell is classified as a time cell, suggesting a strong relationship between time and velocity. The threshold of 0.5 was chosen through an iterative process to achieve a complete separation between the time and distance classifications. This empirical threshold ensures that cells are classified exclusively as either time-related or distance-related, avoiding any overlap or ambiguity in classification.

c) The p-value metric checks the statistical significance of the linear regression, and classifies the cell as time or distance according to the significance of the linearity of distance vs. velocity or time vs. inverse-velocity relations.

While the CellType metric classifies all cells, the FIT and p-value classify only cells within a certain range, resulting in a smaller population of cells. Using all three metrics provides a perspective of the entire population on one hand, while emphasizing a subset of the population with a higher classification confidence level, on the other hand.

All three metrics yielded similar results and had similar statistical power.

The following was added:

“The CellType metric utilizes the variances of the onset times (Ti−T¯)2, where the average onset time is computed across all runs within the session, and of the onset distances (Si−S¯)2, where the average onset distance is calculated across all runs within the session.CellType(Vi,Si,Ti)=ΣiVi∗(Si−S¯)2−Σi(Ti−T¯)2ΣiVi∗(Si−S¯)2+Σi(Ti−T¯)2

CellType is in the range of -1 to 1. For an idealtime encoding cell, the onset variance Σi(Ti−T¯)2 = 0 and hence CellType=1. For an ideal distance encoding cell, the distance variance (multiplied by the respective velocity in order to match units) Vi∗(Si−S¯)2=0, and hence CellType=-1."

Additional metrics were defined and used for classifying the cells encoding:

The “FIT” metric is defined as follows:Fit(m,k,T¯,S¯)={−10.5<mT¯10.5<kS¯0otherwise

Where m and k are the linear fit slope coefficients (from equations 1 and 2), T¯ is the average firing onset time and S¯ is the average distance the animal traveled until the onset. Fit is -1 for a distance cell and 1 for a time cell.

The “P-Value” metric is defined as follows:P_Value(Vi,Si,Ti)={−1p(no linear relation between distance and velocity)<0.051p(no linear relation between the onset and reciprocal velocity)<0.050otherwise

A stricter metric, utilized the statistical significance of the linearity in equations 1 and 2, through F-statistics. We classified a cell as distance encoding if the null hypothesis that there is no linear relation between the distance and velocity was rejected with p<0.05. We classified a cell as time encoding if the null hypothesis that there is no linear relation between the onset and the reciprocal velocity was rejected with p<0.05.” (page 7 line 7)

"To assess the power of the statistics we compared the results to a distribution generated from shuffled trial types (Figure 3c). In order to mitigate any potential biases in this distribution, we truncated all data to a common duration of 16 seconds, which represents the shortest duration of a treadmill run across all trials." (page 4 line 24)

I felt the final paragraph was important but not precise enough and therefore hard to parse.

Paragraph changed as follows:

"Another possible mechanism for acquiring representations that are consistent with task structure involves an associative learning process. Such learning would modify all connections that were active in a particular trial but would only consistently modify those in which activity was invariant throughout the experiment, while other connections associated with inconsistent representations will average out. Thus, in time-fixed experiments, those connections that are consistent with time would be strengthened, while in distance-fixed experiments those that are consistent with distance would gain strength. Consequently, those cells will gradually encode either distance or time, depending on the type of experiment." (page 5 line 21)

The explicit structure of reward and its relationship to the main conjecture is not really discussed. This seems crucial. I'd prefer to have it upfront in the manuscript, rather than mentioning it obliquely at the beginning and coming back to it at the end.

Reward description was added as follows:

"The rats were provided with a small amount of water reward prior to the initiation of the treadmill run and upon its cessation, thereby conditioning them to maintain their snouts positioned at the water port throughout the duration of the treadmill run, and "clamping" their behavior and spatial position" (page 3 line 6)